# In Vivo Application of CRISPR/Cas9 Revealed Implication of *Foxa1* and *Foxp1* in Prostate Cancer Proliferation and Epithelial Plasticity

**DOI:** 10.3390/cancers14184381

**Published:** 2022-09-08

**Authors:** Huiqiang Cai, Simon N. Agersnap, Amalie Sjøgren, Mikkel K. Simonsen, Mathilde S. Blaavand, Ulrikke V. Jensen, Martin K. Thomsen

**Affiliations:** 1Department of Biomedicine, Aarhus University, 8000 Aarhus, Denmark; 2Aarhus Institute of Advanced Studies (AIAS), Aarhus University, 8000 Aarhus, Denmark

**Keywords:** CRISPR, prostate cancer, mouse models, Forkhead box proteins, cell plasticity

## Abstract

**Simple Summary:**

Prostate cancer is diagnosed in one out of eight men, with large implications on life quality. Forkhead box proteins are often found mutated in prostate cancer but their functions are still not fully understood. In this study, we applied CRISPR to investigate the function of two Forkhead box proteins, Foxa1 and Foxp1, in the mouse prostate in combination with loss of Pten. Our results reveal that Foxp1 is a tumor suppressor in prostate cancer progression by controlling proliferation and genes regulated by the androgen receptor. Foxa1 controls cell plasticity, as loss of Foxa1 converted the prostatic luminal cells to basal cells. Hereby, this study sheds light on two distinct functions of Forkhead box proteins in prostate cancer.

**Abstract:**

Prostate cancer is the most common cancer in men in the Western world and the number is rising. Prostate cancer is notoriously heterogeneous, which makes it hard to generate and study in pre-clinical models. The family of Forkhead box (FOX) transcription factors are often altered in prostate cancer with especially high mutation burden in *FOXA1* and *FOXP1*. *FOXA1* harbors loss or gain of function mutations in 8% of prostate cancer, which increases to 14% in metastatic samples. *FOXP1* predominately occurs with loss of function mutations in 7% of primary tumors, and similar incidents are found in metastatic samples. Here, we applied in vivo CRISPR editing, to study the loss of functions of these two FOX transcription factors, in murine prostate in combination with loss of *Pten*. Deficiency of Foxp1 increased proliferation in combination with loss of Pten. In contrast, proliferation was unchanged when androgen was deprived. The expression of *Tmprss2* was increased when Foxp1 was mutated in vivo, showing that Foxp1 is a repressor for this androgen-regulated target. Furthermore, analysis of *FOXP1* and *TMPRSS2* expression in a human prostate cancer data set revealed a negative correlation. Mutation of *Foxa1* in the murine prostate induces cell plasticity to luminal cells. Here, epithelial cells with loss of *Foxa1* were transdifferentiated to cells with expression of the basal markers Ck5 and p63. Interestingly, these cells were located in the lumen and did not co-express Ck8. Overall, this study reveals that loss of *Foxp1* increases cell proliferation, whereas loss of *Foxa1* induces epithelial plasticity in prostate cancer.

## 1. Introduction

Prostate cancer (PCa) is the most common cancer in men in the Western world and the number is rising as life expectancy is increasing [1]. PCa is a slowly developing cancer and remains an indolent disease, but when the cancer has progressed to a metastatic stage, treatment options are limited. As a consequence, metastatic PCa has poor prognosis, with less than 30% overall survival after 5 years [1,2,3]. Targeting the androgen receptor (AR) by medical castration or target therapy results in a regression of the cancer, but eventually cancer cells will gain androgen independence, resulting in a relapse of the cancer [3]. Therefore, understanding the resistant mechanism and the pathways that are dysregulated to drive castration-resistant prostate cancer (CRPCa) progression is crucial for the development of new treatment strategies.

The family of Forkhead box (FOX) transcription factors are often altered in prostate cancer. It is a large gene family that is implicated in multiple biological processes. FOXA1 is required for prostate development and for cell differentiation of the prostatic epithelium [4,5]. FOXA1 facilitates open chromatin conformations and interacts with AR to regulate gene expression [6,7,8]. Hereby, FOXA1 is essential for the homeostasis of the prostate epithelial cells and testosterone-regulated gene program. The function of Foxa1 gain of function mutations has been studied in prostate organoids and revealed oncogenic properties [6,7]. However, loss of function studies of Foxa1 in the mouse prostate induces hyperplasia and shows that Foxa1 can have dual roles in prostate cancer [4]. Another Forkhead box transcription factor, FOXP1, is also implicated in PCa and interacts with AR. Chromatin immunoprecipitation sequence analysis has revealed that FOXP1 binds motifs close to AR binding sites in the promotor regions of different genes [9,10]. The function of FOXP1 in PCa is less understood but is often found lost in the form of chromosomal deletion together with *SHQ1*. A mouse model of this chromosomal deletion has shown transformation of the prostatic epithelium and increased expression of AR-regulated genes [11]. Studies on FOX protein implications in PCa progression have mainly been conducted in cell lines and organ cultures, as the availability of mouse models has been limited. Therefore, studies on FOX protein functions in vivo are still required to better decipher their implications in PCa initiation and progression.

The discovery of CRISPR (clustered regularly interspaced short palindromic repeats)/Cas9 has great implications for cancer research, as it allows for the generation of specific gene mutations [12]. CRISPR inductions of mutations are fast, and it is possible to target multiple genes simultaneously. Furthermore, the method can be applied to in vivo studies by orthotopic viral delivery of multiplexed single guide RNAs (sgRNAs) to Cas9 transgenic mice. This allows targeting multiple genes in specific tissues or cells of interest to generate cells with a unique mutation profile, which can proceed to clonal expansion in a natural setting [12,13,14,15,16]. We applied CRISPR to study cancer in the mouse prostate in vivo, mainly by targeting Pten in combination with other potential tumor-suppressor genes [13,14,17]. Loss of Pten accelerates prostate cancer initiation and is often found mutated in human PCa. Therefore, additional mutation of Pten in the mouse prostate provides a model to study the genes of interest in a transformed prostate [18,19,20,21].

Here, we applied CRISPR to study the function of Foxa1 and Foxp1 deficiency in PCa in vivo. By applying adeno-associated virus (AAV), sgRNAs targeting Foxa1, Foxp1 and Pten were delivered to the murine prostate to initiate PCa. Samples were assessed four months after and revealed that loss of Foxp1 promoted tumor progression by increasing proliferation and expression of Tmprss2. Testosterone ablation by castration impaired the tumor-suppressor function of Foxp1 and revealed Foxp1 as a negative regulator of AR-driven proliferation in the prostatic tissues. Depletion of Foxa1 in combination with loss of Foxp1 and Pten in the murine prostate tissues differentiated cells towards basal cells. Loss of Foxa1 increased expression of p63 and Ck5 and down-regulated Ck8. Overall, we show by CRISPR alteration that Foxp1 is a tumor suppressor by interfering with the AR pathway and Foxa1 is required for luminal cells’ identity.

## 2. Materials and Methods

### 2.1. Animals

B6J.129(B6N)-Gt(ROSA)26Sortm1(CAG-cas9*,-EGFP)Fezh/J mice were purchased from Jackson Laboratories (catalog No. 26175) and bred and housed at Aarhus University. All animal experiments were conducted in accordance with the protocol approved by the Danish Animal Experiments Inspectorate (license no. 2020-15-0201-00711). Housing and care of the mice was in accordance with the Danish animal research proposal on genetically modified animals. Mice were euthanized by cervical dislocation.

### 2.2. sgRNA Design and AAV Vector Constructs

The *Foxa1* and *Foxp1* sgRNA were designed using the (http://crispor.tefor.net, accessed on 17 May 2020) CRISPOR design tool. See Appendix A for sgRNA sequence and genomic primers. The guide for *Pten* has been described before [14]. The guide efficacy was determined by transfection of LSL-Cas9 murine embryonic fibroblasts (MEFs) with pSpCas9-guide-2A-puromycine plasmid (Addgene ID: 48138), and Synthego ICE software was used to assess Indel formation. *Foxa1* and *Foxp1* sgRNAs were cloned to an AAV2 backbone containing *Pten* guide and Cre coding frame [3]. sg*Foxa1* was under the control of a murine U6 promotor (Addgene, 53187) and sg*Foxp1* and sg*Pten* were under a human U6 promotor (Addgene, 62988). The final constructs for AAV productions were as follows: Pten control (AAV:ITR-U6-sgRNA(Pten)-pEFS-Rluc-2A-Cre-ITR), Pten; Foxp1 (AAV:ITR-U6-sgRNA(Pten)-U6-sgRNA(Foxp1)-pEFS-Rluc-2A-Cre-ITR) and Pten; Foxp1; Foxa1 (AAV:ITR-U6-sgRNA(Pten)-U6-sgRNA(Foxp1)-mU6-sgRNA(Foxa1)-pEFS-Rluc-2A-Cre-ITR).

### 2.3. Cell Work and AAV Production

MEF cells from LSL-Cas9 mouse embryos have been described before. MEF and HEK293T cells were grown in DMEM (Sigma-Aldrich, St. Louis, MO, USA), supplemented with penicillin–streptomycin (Sigma-Aldrich) and 10% FCS (Gibco, Waltham, MA, USA) at 37 °C and 5% CO_2_. MEF cells were transfected with X-tremeGENE-9 according to the manufacturer’s protocols and puromycine selections were applied for 48 h (2 µg/mL). AAV production was conducted in HEK293T cells as described before [22].

### 2.4. Virus Delivery to the Prostate and Castrations

Surgical delivery of AAV to the murine prostate was performed according to the previously established protocol [17]. In brief, 10-week-old mice were anesthetized with a mixture of medetomidin (40 mg/kg), midazolam (15 mg/kg), and burophanol (10 mg/kg), and an incision of 1.2 cm was made in the abdomen above the bladder. The seminal vesicles were lifted out and 20 µL virus (10^9^ viral genomes) was injected into the anterior prostate lobe at each side. The tumor progression proceeded for 4 months, and the experiment was terminated afterwards. A subset of mice underwent castration 3 months post-operation and were left untreated for another month before the prostate was collected. For castration, the mice received anesthetization and an incision was made in the abdomen and the testes were lifted out. The blood supply was ligated and the testes were removed before the abdomen was closed.

### 2.5. Histochemical Analysis

Tissue samples were fixed in 4% paraformaldehyde (Santa Cruz Biotechnology, Dallas, TX, USA) before being dehydrated and embedded in paraffin. Tissue sections of 4 µm were cut and underwent deparaffinization before antigen retrieval was performed at 100 °C in a citrate buffer at pH 6. Sections were blocked in 2.5% BSA (Sigma-Aldrich) in PBST prior to incubation with the following primary antibodies: pAkt (CST-4060), Ki67 (MA5-14520), Foxp1 (CST-4402), Foxa1 (NBP2-45354), p63 (CST-39692), Ck5 (BioLegend, 905501, San Diego, CA, USA), androgen receptor (Merck, 06-680, Darmstadt, Germany) or Ck8 (BioLegend, 904801). Appropriate horseradish-peroxidase-conjugated (Vector HRP) or florescent secondary antibodies were used (Invitrogen, Waltham, MA, USA). Counterstaining was performed with hematoxylin or DAPI.

### 2.6. DNA/RNA Isolation and PCR

DNA and RNA were isolated from the prostate tissue with the Qiagen AllPrep DNA/RNA kit according to the manufacturer’s protocols (Qiagene, Hilden, Germany). PCR was performed on genomic DNA with Dream Taq master mix (Thermo Scientific, Waltham, MA, USA). See Appendix A for primers. Q-PCRs for target genes were performed on 20 ng total RNA from tissue samples with Brilliant III Ultra-Fast SYBR^®^ Green QPCR Master Mix (Agilent Technologies, Glostrup, Denmark) (see Appendix A for primer sequences). Data were analysed with the ΔΔCT method and normalized to *Rpl32*.

### 2.7. Statistics

GraphPad Prism was used for statistics analysis. An unpaired t-test was used for Ki67 quantification and analysis of gene expression. A *p*-value ≤ 0.05 was considered statistically significant between the two groups.

## 3. Results

### 3.1. FOXA1 and FOXP1 Are Commonly Mutated in PCa

*FOX* genes have been reported to be among the most commonly mutated genes in PCa [23]. To assess if the mutation frequency was different in primary and metastatic PCa, two publicly available data sets were analysed. The mutation frequencies of *FOXA1* and *FOXP1* were assessed in primary PCa from the TCGA (Firehose legacy; https://www.cancer.gov/tcga, accessed on 5 February 2020) data set and revealed mutations in 8 and 7%, respectively (Figure 1). In metastatic PCa, *FOXA1* was found mutated in 14% of the samples, but for *FOXP1* the number was not increased (SU2C/PCF [24]) (Figure 1). Furthermore, the analysis indicated that the mutations in these two *FOX* genes were mutually exclusive in primary PCa but co-occurrent in metastatic disease (*p* = 0.029). Phosphatase and tensin homolog (PTEN) is another commonly mutated gene in PCa, and is a negative regulator of the PI3K pathway [20]. To assess the mutation frequency of *PTEN* in relation to *FOXA1* and *FOXP1*, the same two data sets were analysed. *PTEN* was mutated in 22% of primary PCa, and this increased to 33% in metastatic samples (Appendix A). The loss of *PTEN* in primary PCa co-occurred with a loss of *FOXP1* (*p* = 0.028), but not with *FOXA1* alterations. Interestingly, mutations in *FOXA1* were predicted to be either gain or loss of function, indicating that *FOXA1* can act as an oncogene or tumor suppressor in PCa. In contrast, mutations in *FOXP1* and *PTEN* often resulted in loss of function (Figure 1 and Appendix A). These data revealed that *FOXA1* and *FOXP1* have distinct functions in PCa. *FOXP1* is likely a tumor-suppressor gene in primary PCa, whereas the function of *FOXA1* is context-specific.

### 3.2. Loss of Foxp1 and Pten by CRISPR in the Murine Prostate

To investigate the function of Foxp1 in PCa initiation and progression in vivo, we applied CRISPR to generate loss of function. CRISPR guides for *Foxp1* were designed to target the 5′ end of the coding sequence to induce insertion or deletions (indel) in the reading frame. Guide RNAs for *Foxp1* were cloned into a plasmid containing Cas9 and puromycin coding genes. The guide efficiency was validated in mouse embryonic fibroblasts (MEFs) and the guide with the highest efficiency was selected for further use (Appendix A).

The selected guide RNA for *Foxp1* was cloned into an AAV backbone together with a specific sgRNA for *Pten* (Figure 2A). The AAV backbone also provided expression of Cre protein for activation of Cas9 and EGFP expression in transduced cells from transgene Cas9-EGFP mice [16]. AAVs were produced and MEF cells were transduced for a validation of the virus. Transduced MEF cells started to express EGFP due to Cre-mediated removal of a stop codon, confirming appropriated virus activation (Figure 2B). DNA was isolated from the MEF cells and Sanger Sequencing was performed on the target regions of the two sgRNAs. The formation of indels was detected for both guides, confirming that the AAVs were capable of transducing cells and introducing indel formations at the designated targets (Figure 2C).

To induce PCa, ten-week-old Cas9 transgene mice were injected with AAV in the anterior prostate lobe. Loss of *Foxp1/Shq1* in the mouse prostate has previously shown not to induce PCa [11]. Therefore, mice were injected with AAV particles that contain guide RNAs for *Foxp1* and *Pten*, whereas control mice received AAV particles that only contained a guide for *Pten*. Four months post-virus delivery, the experiments were terminated and the prostates were isolated. The anterior lobes were enlarged in both groups of mice and areas with EGFP expression were visible (Figure 2D). Next, the isolated prostate samples were analysed by histological alteration. H & E staining revealed areas of high-grade prostatic intraepithelial neoplasia (PIN) [25,26] in the anterior lobe and alteration in the stroma (Figure 2E). These are all features related to loss of Pten and activation of PI3K in the prostatic epithelium. Immunohistochemistry (IHC) staining for phosphorylated Akt (p-Akt) confirmed increased levels in the transformed areas, and staining for Pten revealed loss of expression in both groups of mice (Figure 2E and Appendix A). Staining for Foxp1 revealed loss of expression in the transduced area, whereas mice receiving only *Pten* virus showed strong expression of Foxp1. Staining for p63, a marker of basal cells, showed similar expression in both groups as was also seen for AR expression (Figure 2E and Appendix A). DNA was isolated from the prostatic tissues, and Sanger Sequencing was conducted to analyze the indels at the target genes. *Pten* was mutated in the control samples and in the mice that had been targeted by two guides. Analysis of *Foxp1* alteration showed similar indel frequency as for *Pten*, indicating that loss of both genes co-occurs with the delivery of both guide RNAs (Figure 2F). Overall, by in vivo application of CRISPR/Cas9 gene editing, *Pten* and *Foxp1* were mutated in the mouse prostatic epithelium.

### 3.3. Loss of Foxp1 Increases Androgen-Dependent Proliferation in the Mouse Prostate

The function of FOXP1 in PCa has been associated with tumor-suppressor functions through in vitro studies [10]. To assess the function in vivo, mice with loss of *Foxp1* and *Pten* in the prostate epithelium were compared to *Pten*-deficient animals. IHC was performed on tissue sections for Foxp1 and p-Akt to identify areas with loss of both Pten and Foxp1. Hereafter, the tissue sections were stained for the proliferation marker Ki67, and positive cells were counted. Loss of *Foxp1* in combination with *Pten* deficiency increases proliferation compared to the control with only loss of *Pten* (Figure 3A,B). Foxp1 has been suggested to be a negative regulator of the AR transcriptional network. The expression of AR-regulated genes was assessed in the murine samples. Four genes, of which expression has been associated with AR, were assessed but only *Tmprss2* was found to be affected. *Tmprss2* expression was increased 3.5-fold when *Foxp1* was lost, whereas *Nkx3.1*, *App* and *Klk4* were unaltered (Figure 3C). As androgen is a positive regulator of cell proliferation in the prostatic tissues, the increased proliferation in *Foxp1*-deficient tissues could be a consequence of increased AR activity. To assess the proliferation in absence of AR regulation, mice underwent castration 3 months after cancer initiation, and the prostatic tissues were analysed one month post-castration. Histological H & E and AR staining revealed that androgen had been abolished, and the samples were stained for Foxp1 and p-Akt to confirm CRISPR-induced mutations (Figure 3D, Appendix A). Hereafter, sections were stained for Ki67 and the positive cells were counted from the two groups. The results showed that the increase of cell proliferation seen in *Foxp1*-deficient prostate was dismissed in the absence of testosterone (Figure 3E). These data show that Foxp1 is a negative regulator of proliferation in the presence of testosterone in PCa. The in vivo analysis showed a negative correlation between *Foxp1* expression and the AR-regulated gene *Tmprss2*. To evaluate if *FOXP1* and *TMPRSS2* are negatively correlated in human prostate cancer, TCGA data sets were analysed. The expressions of *FOXP1* and *TMPRSS2* are negatively correlated in prostate adenocarcinoma (TCGA, PanCancer Atlas; https://www.cancer.gov/tcga, accessed on 9 May 2022).

However, expressions of *NKX3-1*, *APP* and *KLK4* were not negatively correlated to *FOXP1*, similarly to the observations from our mouse model above (Figure 3F). These analyses show a consistency between mice and humans concerning FOXP1 regulations of *TMPRSS2* expression.

### 3.4. Foxa1 Regulates Prostatic Cell Plasticity

*FOXA1* is often altered in PCa as loss of function mutations or gain of function through amplifications. To assess the role of *Foxa1* loss in PCa, we cloned a *Foxa1* sgRNA for our AAV construct. Hereby, the construct contained sgRNAs for *Pten*, *Foxa1* and *Foxp1* and Cre expression (Figure 4A). AAV particles were produced and MEF cells were transduced for validation of the virus. The MEF cells showed expression of EGFP, indicating Cre expression from the AAV and removal of the stop codon in front of Cas9 and EGFP (Figure 4B). Indel analysis by Sanger Sequencing confirmed mutations induced by all three guides, guaranteeing the in vivo application of the virus (Figure 4C). To assess the function of *Foxa1* deficiency in combination with loss of *Pten* and *Foxp1* in PCa, ten-week-old Cas9 transgene mice were injected with AAV in the anterior prostate lobe. Four months after, the prostates were collected for further analysis. The anterior lobe was enlarged for triple deficient mice but the sizes were similar to mice with only loss of *Pten*. Both groups of mice contained areas with EGFP expression in the prostatic lobes, confirming Cre recombination activities from the AAV (Figure 4D). Histological analysis showed features of high-grade PIN in the altered areas in the anterior prostate (Figure 4E). Staining for p-Akt showed increased levels in both genotypes, indicating a loss of Pten.

Staining for Foxa1 and Foxp1 showed a loss of expression in triple deficient samples but a high level of expression in the Pten^ΔP^ control (Figure 4E). GFP positive biopsies were dissected for DNA and RNA extraction. Sanger Sequencing for CRISPR-induced mutations showed indel in all samples for *Pten*, *Foxa1* and *Foxp1* (Figure 4F). Foxa1 has been shown to regulate AR target genes, as seen for Foxp1 [4,21]. Therefore, qPCR was performed for the expression of the four targets, *Tmprss2*, *Nkx3.1*, *App* and *Klk4*. *Tmprss2* mRNA was increased as had been observed with the loss of Foxp1. However, *App* mRNA was also upregulated, indicating that Foxa1 is a possible negative regulator of this gene in vivo (Figure 4G).

The indel frequencies for *Foxp1* and *Pten* were significantly increased, compared to the indel for *Foxa1*, even though the same samples were analysed (Appendix A). Therefore, immunofluorescence staining for Foxa1 and p-Akt was performed, and it revealed cells positive for both Foxa1 and p-Akt. This shows sub-areas with loss of Pten, although without loss of Foxa1 in the altered prostatic tissues, suggesting a negative selection for loss of Foxa1 in this context (Figure 5A,B).

Next, samples were stained for the basal cell marker p63, which showed normal basal layer at the stroma. Interestingly, triple deficient samples displayed p63 positive cells in the lumen. These were not seen in Pten^ΔP^ control samples or samples deficient in *Pten* and *Foxp1* (Figure 2E and Figure 5C). To evaluate if loss of *Foxa1* may induce basal cell marker expression, samples were co-stained for Foxa1 and the basal marker Ck5. Staining revealed that Ck5-expressing cells in the lumen were negative for Foxa1 (Figure 5D). This indicates a negative correlation between Foxa1 and expression of basal cell markers. Ck5 positive cells in the lumen of the prostate have been shown to also express Ck8 and these cells have been marked as trans-amplifying cells [18,21]. To assess if loss of Foxa1 drives luminal cells into trans-amplifying cells, co-staining for Ck5 and Ck8 was conducted on tissue sections. Interestingly, Ck5-expressing cells did not co-express Ck8 in the prostatic lumen and no differences in trans-amplifying cells were observed (Figure 5E,F). Overall, the loss of Foxa1 drives cell plasticity to induce basal cell expression in the luminal space of the prostatic lobe.

## 4. Discussion

In this study, we analysed the function of two Forkhead box protein functions in PCa initiation and progression in vivo. We applied CRISPR to generate loss of function of Foxa1 and Foxp1 in the murine prostate, in combination with the activated Pi3K pathway by simultaneous depletions of *Pten*. Our results show that Foxa1 and Foxp1 have distinct functions in PCa, as Foxa1 regulates cell plasticity and Foxp1 controls cell proliferation. Furthermore, both Fox proteins regulate specific gene expressions in vivo, which are known as AR targets. These findings are coherent with expression data from human PCa samples. Overall, we have shown that *Foxa1* and *Foxp1* are essential genes in prostate biology.

The implications of Fox proteins in prostate biology have been studied for many years. In particular, FOXA1 has been investigated thoroughly in prostate development [5] and in PCa, as this transcription factor commonly harbors genetic alterations in PCa [4,6]. Here, we analysed the mutation profile of *FOXA1* in the TCGA data set containing more than 500 samples. This revealed that 8% of primary PCa shows alterations in *FOXA1*, as either loss or gain of function mutations. We focused on *Foxa1* loss of function mutations in this study and revealed that *Foxa1* is crucial for the cell identity. Loss of *Foxa1* in the prostatic epithelium regulates cell plasticity, as the cells express basal cell markers even though the cells are located in the lumen of the prostatic lobe. The location of cells expressing basal markers in the lumen could involve invading of basal cells but we have not seen any evidence of this. Furthermore, immunofluorescence staining shows that Foxa1 expression is restricted to luminal cells in the prostatic lobe, indicating that Foxa1 is essential for luminal cell lineage. Foxa1 implication in cell identity has been associated with luminal cell expansion in organ culture studies and decrease of luminal cells in vivo [4,6,7]. This indicates that loss of Foxa1 is responsible for this change of cell identity, even though it has been studied in combination with loss of *Foxp1* and *Pten*. Here our data show that loss of *Foxa1* in adult prostate tissues differentiates the cells into basal-like cells with loss of luminal signatures.

FOXA1 and FOXP1 have been shown to bind genomic regions containing overlapping AR binding sites and hereby synergistically regulate AR target genes in vitro [9,10,27]. We investigated the expression of four established AR targets, *Nkx3.1*, *App*, *Klk4* and *Tmprss2* in the murine prostate after loss of Foxp1. Interestingly, only *Tmprss2* expression increased, whereas *Nkx3.1*, *App* and *Klk4* expressions were unchanged. This shows that Foxp1 is working as a negative regulator on selected AR-regulated *genes* in vivo, and similar results were obtained from gene expression data sets of human PCa [11]. AR is known to increase cell survival and proliferation of the prostatic tissues, and we showed that loss of Foxp1 increased proliferation in the presence of testosterone together with an effect on *Tmprss2* expression. This indicated that Foxp1 is a negative regulator of AR target genes that are responsible for proliferation. Consistent with our findings, others have shown upregulation of AR-regulated genes and increased proliferation when Foxp1 is depleted [10,11]. Future work will focus on Foxp1 molecular interaction on AR-regulated genes to shed light on the underlying mechanisms in PCa.

Loss of *Foxa1* in combination with an abrogation of *Foxp1* and *Pten* showed increased expression in *App*. This suggests that in vivo, Foxa1 is a negative regulator of *App.* Whether this dysregulation of *App* is directly controlled by Foxa1 or a combination of loss of both Fox genes has not been assessed. App has been shown to promote PCa growth and cell migration [27]. Hereby, *Foxa1* can act as a tumor-suppressor gene in the prostatic epithelium, even though the majority of genetic alterations in *FOXA1* are found as gain of function mutations [6]. Future work will apply CRISPR technology to study gain of function mutations of Foxa1 in the murine prostate.

In this study, we applied a CRISPR/Cas9 mouse model to investigate *Foxa1* and *Foxp1* in vivo. With this method, multiple genes can be simultaneously altered in the prostatic epithelial cells of the adult male mouse. Furthermore, tumorigenesis is initiated in single cells, which allows clonal expansion under natural selection, while empty vector or non-targeting sgRNA does not cause PCa formation [13,14]. Here, we observed that *Foxa1* was found mutated to a lesser extent than *Foxp1* and *Pten*. In contrast, indel formation in the MEF cells showed the same efficiency for Fox gene sgRNAs, whereas *Pten* had a decreased mutation rate. We have previously observed that the *Pten* sgRNA had a decreased efficiency in vitro, but was always found mutated in the transformed prostatic epithelium in vivo [14]. In this study, Pten was found lost in all PIN formations, showing a strong selection for mutations in this gene for the initiation of PCa formation. However, *Foxa1* was absent in sub-areas of PIN formations, suggesting a negative selection for loss of *Foxa1* in this context. This could be a biological selection, as loss of Foxa1 differentiated the luminal cells into basal cells. However, it could also reflect technical issues, as *Foxa1* sgRNA was under the murine U6 promotor, whereas *Pten* and *Foxp1* sgRNAs were controlled by human U6. Future work will investigate this possibility, but the in vitro data did not suggest differences in promotor activity, and other studies have also shown similarities between these Pol III promotors [28]. Overall, our CRISPR/Cas9 model for studying gene function in PCa provided new knowledge concerning the function of Foxa1 and Foxp1. The model allowed depletions of these factors in adult prostate epithelium in combination with a loss of Pten.

## 5. Conclusions

In this study, the functions of *Foxa1* and *Foxp1* in PCa have been assessed in vivo in combination with *Pten* mutation. By applying CRISPR/Cas9, mutations have been introduced to the adult mouse prostate and it was revealed that *Foxp1* acts as a tumor-suppressor gene by controlling proliferation in an androgen-dependent manner. Molecular analysis revealed that Foxp1 is a negative regulator of a specific AR-regulated gene, Tmprss2, showing that Foxp1 is essential for prostate biology.

Depletion of *Foxa1* in the context of *Foxp1* and *Pten* mutations differentiated the prostatic epithelial cells to basal cells. This showed that *Foxa1* is essential for prostate luminal cell identity. Furthermore, loss of *Foxa1* also increased the expression of *App*, which is an AR-regulated gene, showing that Foxa1 can alter the expression of testosterone-regulated genes in the prostate. Overall, this study shows that two commonly mutated *FOX* genes in human PCa have distinct functions when studying in an in vivo setting, underlining the complexity of PCa initiation and progression.

## Figures and Tables

**Figure 1 cancers-14-04381-f001:**
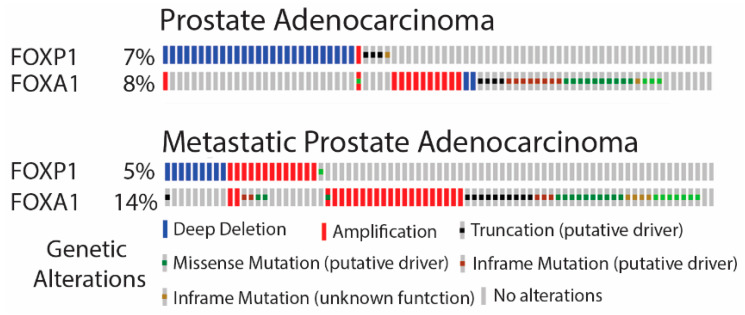
*FOXA1* and *FOXP1* are highly mutated in primary and metastatic prostate adenocarcinoma. The mutation frequency of *FOXA1* and *FOXP1* in primary PCa was assessed in a TCGA data set containing 501 samples (Firehose Legacy). The mutation burden in metastatic prostate cancer was assessed in the SU2C/PCF Dream Team data set containing 444 samples. Data were generated from cBioPortal.org (accessed on 5 February 2020).

**Figure 2 cancers-14-04381-f002:**
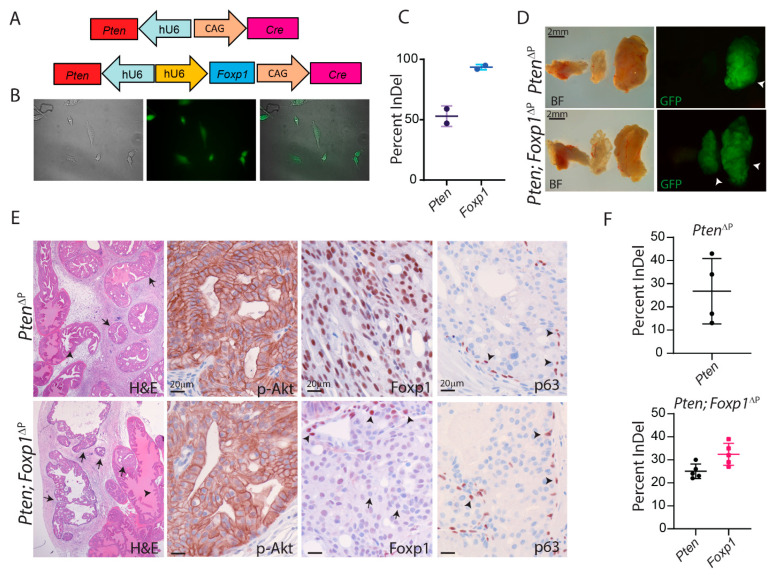
CRISPR generated loss of Foxp1 and Pten in the murine prostate. (**A**) AAV constructs for expression of sgRNAs to *Pten* or *Pten* and *Foxp1* together with expression of Cre protein were generated. (**B**) Validation of the AAV particles were conducted in MEF isolated from Cas9-EGFP^flox/flox^ mice. The expression of EGFP revealed viral transduction and Cre expression from the viral construct. Images were taken with 10× magnification (**C**) The efficiency of the sgRNA was determined with Sanger Sequencing and analysed with Synthego ICE software. (**D**) The AAV particles were delivered to the murine prostate with surgical injections and prostate tissues were examined 4 months post-treatment. White arrowheads mark transformed areas in the prostatic tissues (representative image, *n* = 5). (**E**) Paraffin sections from the prostates were stained for H & E (arrows mark PIN and arrowheads mark normal prostatic epithelial), p-Akt, Foxp1 and p63 (brown stain) to confirm PIN formation and disruption to the Pten pathway and loss of *Foxp1* in the Cas9-EGFP transgene mice (*n* = 5). Arrows mark cells depleted for Foxp1 and arrowheads mark cells positive for Foxp1 or p63. (**F**) Indel frequency was measured on EGFP-positive biopics from the anterior prostate lobe. Sanger Sequencing data were analysed with Synthego ICE software (*n* = 4–5).

**Figure 3 cancers-14-04381-f003:**
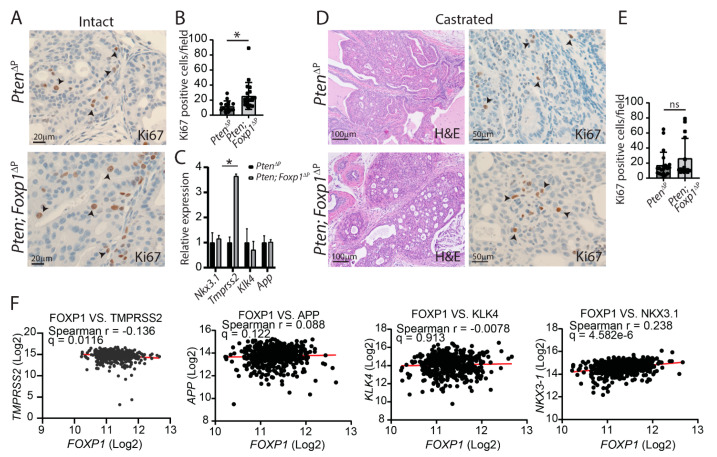
Loss of Foxp1 increases proliferation and expression of *Tmprss2* in the prostate in an androgen manner. (**A**) Tissue sections from the anterior prostate lobes from *Pten* and *Pten; Foxp1* deficient samples were stained for Ki67 (*n* = 5). (**B**) Quantification of Ki67-positive cells was assessed from a total of 25 fields from five mice (* = *p* < 0.05). (**C**) Expressions of AR-regulated genes were measured from *Pten* and *Pten*; *Foxp1*-deficient samples (*n* = 4, * = *p* < 0.05). (**D**) Mice deficient in *Pten* or *Pten; Foxp1* in the prostatic tissues underwent castration one month before samples were collected. Tissue sections were stained with H & E and Ki67. Black arrowheads mark positive cells (brown stain) (*n* = 5). (**E**) Ki67-positive cells were quantified from both genotypes from a total of 15 fields (ns = non-significant). (**F**) The correlation between *FOXP1* and AR-regulated genes was assessed from the TCGA Pan Cancer data set containing 494 samples.

**Figure 4 cancers-14-04381-f004:**
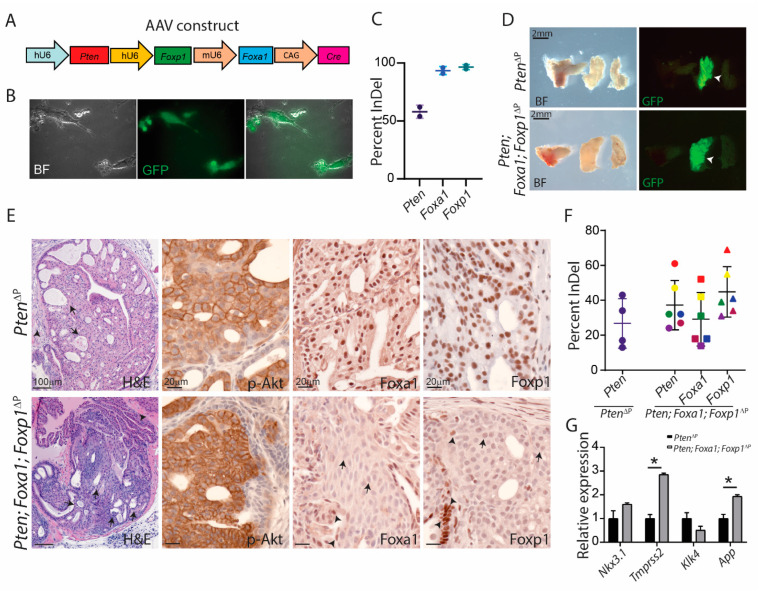
CRISPR-mediated mutations of Foxa1, Foxp1 and Pten in the prostatic epithelium. (**A**) AAV construct for delivery of sgRNAs targeting *Pten*, *Foxa1* and *Foxp1* under pol3 promoters. The construct also contained Cre expression by the CAG promotor. (**B**) AAV particles were validated in MEF from Cas9-EGFP^flox/flox^ mice and expression of EGFP could be detected after viral transduction. Images were taken with 10× magnification (**C**) Indel analysis of the target site from the transduced MEF cells (*n* = 2). (**D**) Macroscopic pictures of the prostatic lobes 4 months after virus transduction. White arrowheads mark the anterior prostate lobes positive for EGFP expression after virus-mediated Cre expression (representative image, *n* = 5). (**E**) Paraffin sections from the prostates were stained for H & E (arrowheads mark PIN and arrows mark stromal changes) and antibodies for p-Akt, Foxa1 and Foxp1 (brown stain) (*n* = 5). Arrows mark cells depleted for Foxa1 or Foxp1, whereas arrowheads mark positive cells. (**F**) Indel frequency for *Pten*, *Foxa1* and *Foxp1* were measured from EGFP-positive biopsies taken from the anterior prostate lobe. Sanger Sequencing data were analysed with Synthego ICE software (*n* = 4–6). (**G**) Expressions of AR-regulated genes were measured from *Pten* and *Pten; Foxa1; Foxp1*-deficient samples (*n* = 4, * = *p* < 0.05).

**Figure 5 cancers-14-04381-f005:**
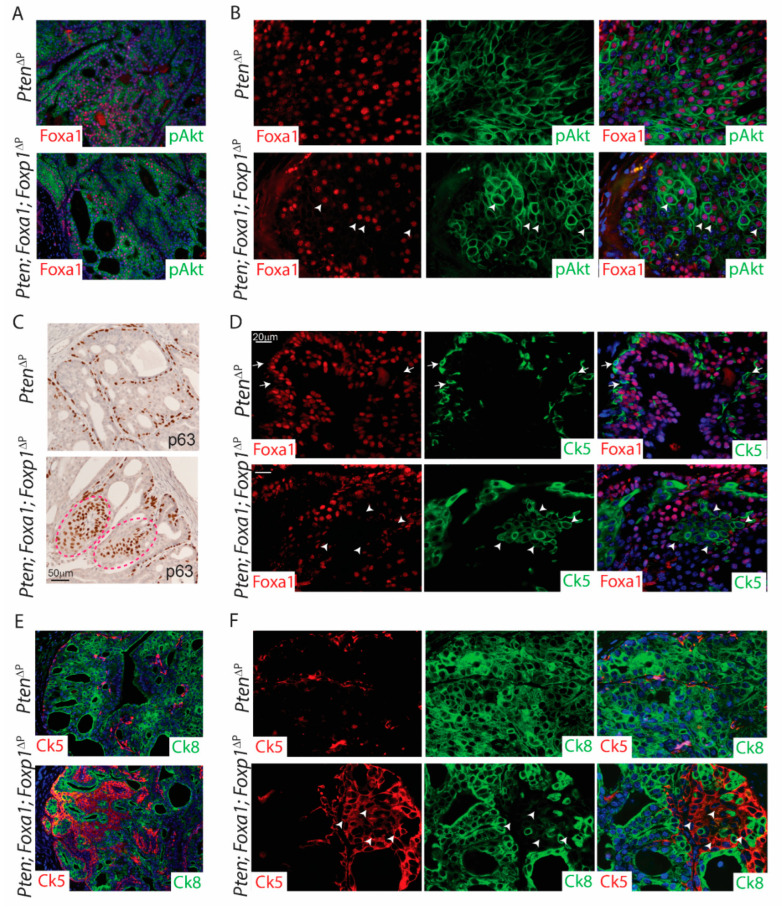
Loss of Foxa1 drives prostatic cell-plasticity. Paraffin sections from *Pten*^Δ*P*^ and *Pten; Foxa1; Foxp1*^Δ*P*^ prostates were stained with antibodies. (**A**,**B**) Immunofluorescence staining for p-Akt (green) and Foxa1 (red). White arrowheads mark Foxa1 and pAkt double positive cells (*n* = 3). (**C**) IHC staining for basal cell marker p63. Red dotted line marks abnormal areas positive for basal cells in the prostatic lumen (*n* = 5). (**D**) Immunofluorescence staining for Ck5 (basal marker; green) and Foxa1 (red). White arrows mark basal cells towards the stroma in Pten-deficient samples. White arrowheads marks Ck5-positive and Foxa1-negative cells in the lumen of Pten; Foxa1; Foxp1-deficient anterior prostate lobes (*n* = 3). (**E**,**F**) Immunofluorescence staining for Ck5 (basal marker; red) and Ck8 (luminal marker; green). White arrowheads mark Ck5-positive and Ck8-negative cells in the prostatic lumen (*n* = 3). Panels on the left are taken with 10× magnification and panels on the right are taken with 40× magnification.

## Data Availability

The data presented in this study are available in this article and Appendix A.

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
