# Peer review of "In Vivo Application of CRISPR/Cas9 Revealed Implication of Foxa1 and Foxp1 in Prostate Cancer Proliferation and Epithelial Plasticity"

_cancers, 2022, doi:10.3390/cancers14184381_

Round 1

Reviewer 1 Report

General comments:

The authors present a manuscript describing a study addressing the role of two Forkhead box proteins, Foxa1 and Foxp1, in prostate cancer. Foxa1 is an established pioneer factor that induce opening of the chromatin to facilitate binding of other transcription factors with which it regulates transcription. Foxa1 is a critical regulator of prostate development. Foxa1 mutations are common in PCa and it is known to contribute to PCa risk particularly by co-operating with androgen receptor (AR) signaling. The role of Foxp1 is less well defined as it has been suggested to function as tumor suppressor in PCa by inhibiting AR signaling but other studies have proposed a role as an oncogenic factor. New insight into the mechanistic functions of the forkhead box proteins would certainly be valuable to the cancer research field. Here, the experimental setup is based on a transgenic mouse model carrying Cre-regulatable bicistronic expression of Cas9 and eGFP where specific gene inactivation can be induced by transducing target cells with viral vectors (AAV was used in this study) carrying gRNA and Cre-expression cassettes. The authors demonstrate the utility of the study system and report that simultaneous depletion of PTEN, Foxp1 and Foxa1 results in high grade PIN lesions with elevated AR-signaling activity. Overall, while most of the findings are not really novel, this is a well-written study with some interesting results and describing a versatile model system to study PCa. While the data appears to be of high quality and the model system is clearly insightful, there are some controls missing which complicates the interpretation of the data. These issues are detailed below and should be clarified to better support the conclusions drawn from the data in this study.  

Specific comments:

1. The authors report that simultaneous depletion of PTEN and Foxp1 resulted in modest increase in proliferation and selective upregulation of one AR-target gene, TMPRSS2. The supporting data on correlated expression in the TCGA Pan Cancer data set does not seem to be very strong. In contrast, three out of four analyzed AR-target genes were upregulated upon co-depletion of Foxa1 together with PTEN and Foxp1 induced. The author conclude that Foxa1 is a negative regulator of App and thereby tumor suppressor in this setup. However, the authors also show that Foxa1 expression tends to be retained in selected lesions of the triple-negative hgPIN tissues essentially suggesting the opposite conclusion. And how can the authors exclude that in this context it is the loss of Foxp1 rather than loss of Foxa1 that accounts for upregulation of the AR target genes? Does Foxa1 depletion in the PTEN-negative background give similar results or is this dependent on loss of both Foxa1 and Foxp1? The authors may need to reconsider their conclusions or provide further supporting evidence to pinpoint the details underlying the potential negative transcriptional regulation by Foxa1 (and Foxp1).

2. As mentioned above, Foxa1-deletion in the triple-KO cocktail appears to be selected against in vivo. Interestingly, the triple negative hgPINs are shown to contain cells clusters with basal characteristics. Are these cells foxa1-positive or -negative? It seems that authors propose that these cells are Foxa1-negative - but are they? What is the evidence supporting the proposed conclusion that Foxa1 is essential for luminal differentiation? Do the authors know whether the phenotype results from expansion of basal population or (de)differentiation of luminal cells into basal ones? It should be noted that Foxa1 has already been shown to be responsible of the maintenance of differentiated phenotype in the prostate and its loss results in expansion of cells with basal characteristics (references 4and5; these papers should be discussed in more detail in the discussion section as they are directly related to the findings of the current study; reference 6 is also relevant showing luminal differentiation but in response to expression of mutated Foxa1). This result is perhaps the most intriguing finding of the study that opens up many possibilities to address questions also related to tumor microenvironmental growth and differentiation control, in addition to cell plasticity questions.

3. The claim that Foxp1 is a negative regulator of AR target genes regulating proliferation seems to be an overstatement in light of the current available data in this study. If this statement also refers to data from earlier similar studies such as reference 11 then it should be elaborated in more detail.

4. Why were these four AR-target genes selected? Are both Foxa1 and Foxp1 known to bind to their promoters? How are these results in line with the data in reference 11 where it appears that more AR target genes are affected?

Author Response

We thank the reviewers for the thorough and constructive comments on our manuscript. We have addressed the majority of the questions, which has mainly been clarification and modification of data interpretations. We have added a new supplementary figure containing IHC staining for androgen receptor, as requested by reviewer 2.

We have tried to address all the questions but if any reviewer feels that this has been insufficient, we will happily elaborate on the matter.

For the reviewers information, the text has been edited and moved in the manuscript document. This has resulted in figure texts moved and not standing below the figures. We have not edited this, as the file is with track changes and this would be confusing.

General comments:

The authors present a manuscript describing a study addressing the role of two Forkhead box proteins, Foxa1 and Foxp1, in prostate cancer. Foxa1 is an established pioneer factor that induce opening of the chromatin to facilitate binding of other transcription factors with which it regulates transcription. Foxa1 is a critical regulator of prostate development. Foxa1 mutations are common in PCa and it is known to contribute to PCa risk particularly by co-operating with androgen receptor (AR) signaling. The role of Foxp1 is less well defined as it has been suggested to function as tumor suppressor in PCa by inhibiting AR signaling but other studies have proposed a role as an oncogenic factor. New insight into the mechanistic functions of the forkhead box proteins would certainly be valuable to the cancer research field. Here, the experimental setup is based on a transgenic mouse model carrying Cre-regulatable bicistronic expression of Cas9 and eGFP where specific gene inactivation can be induced by transducing target cells with viral vectors (AAV was used in this study) carrying gRNA and Cre-expression cassettes. The authors demonstrate the utility of the study system and report that simultaneous depletion of PTEN, Foxp1 and Foxa1 results in high grade PIN lesions with elevated AR-signaling activity. Overall, while most of the findings are not really novel, this is a well-written study with some interesting results and describing a versatile model system to study PCa. While the data appears to be of high quality and the model system is clearly insightful, there are some controls missing which complicates the interpretation of the data. These issues are detailed below and should be clarified to better support the conclusions drawn from the data in this study.  

Specific comments:

  1. The authors report that simultaneous depletion of PTEN and Foxp1 resulted in modest increase in proliferation and selective upregulation of one AR-target gene, TMPRSS2. The supporting data on correlated expression in the TCGA Pan Cancer data set does not seem to be very strong. In contrast, three out of four analyzed AR-target genes were upregulated upon co-depletion of Foxa1 together with PTEN and Foxp1 induced. The author conclude that Foxa1 is a negative regulator of Appand thereby tumor suppressor in this setup. However, the authors also show that Foxa1 expression tends to be retained in selected lesions of the triple-negative hgPIN tissues essentially suggesting the opposite conclusion. And how can the authors exclude that in this context it is the loss of Foxp1 rather than loss of Foxa1 that accounts for upregulation of the AR target genes? Does Foxa1 depletion in the PTEN-negative background give similar results or is this dependent on loss of both Foxa1 and Foxp1? The authors may need to reconsider their conclusions or provide further supporting evidence to pinpoint the details underlying the potential negative transcriptional regulation by Foxa1 (and Foxp1).

Response:

The upregulation of App is not seen in samples with loss of Foxp1 and Pten or only Pten. This is only seen in samples where all three genes are depleted. As we have not addressed the App expression in Foxa1 and Pten deficient samples but only in triple KO prostate, we cannot rule out that the increased expression of App is a consequence of loss of both Foxp1 and Foxa1 and not only Foxa1. We have now modified the text on line 323 and the discussion on line 398-402, to highlight this possibility.

  1. As mentioned above, Foxa1-deletion in the triple-KO cocktail appears to be selected against in vivo. Interestingly, the triple negative hgPINs are shown to contain cells clusters with basal characteristics. Are these cells foxa1-positive or -negative? It seems that authors propose that these cells are Foxa1-negative - but are they? What is the evidence supporting the proposed conclusion that Foxa1 is essential for luminal differentiation? Do the authors know whether the phenotype results from expansion of basal population or (de)differentiation of luminal cells into basal ones? It should be noted that Foxa1 has already been shown to be responsible of the maintenance of differentiated phenotype in the prostate and its loss results in expansion of cells with basal characteristics (references 4and5; these papers should be discussed in more detail in the discussion section as they are directly related to the findings of the current study; reference 6 is also relevant showing luminal differentiation but in response to expression of mutated Foxa1). This result is perhaps the most intriguing finding of the study that opens up many possibilities to address questions also related to tumor microenvironmental growth and differentiation control, in addition to cell plasticity questions.

Response:

The reviewer is raising an interesting question. One advantage/ disadvantages of CRISPR induced mutations compared to conditional KO models is that clones with different mutation profile can appear. We have in this study observed lower mutation rate for Foxa1 compared to Foxp1 and Pten by DNA sequencing (Figure 4f). We have also identified areas in the prostatic lumen with activated pAkt (presumable due to loss of Pten) but with cells staining positive for Foxa1 (figure 5a-b). This suggests that loss of Pten has occurred without Foxa1 having been mutated. Similarly, we have co-stained for CK5 (basal cell marker) and Foxa1, and found a negative correlation (figure 5d). Especially the positive basal cells in the lumen are staining negative for Foxa1. We have discussed this but as the reviewer points out, we cannot rule out that and expansion of basal cells has occurred. However, we have not seen any increase in trans-amplifying cells (CK5 and CK8 positive), which could indicate a migration of basal cells. Instead, we suggest that loss of Foxa1 has allowed luminal cells to differentiate into basal cells by upregulation of CK5/p63 and downregulation of CK8. We have included this in the discussion and made reference to related literature, which has also indicated Foxa1 to be essential for luminal cells. See line 372-383

  1. The claim that Foxp1 is a negative regulator of AR target genes regulating proliferation seems to be an overstatement in light of the current available data in this study. If this statement also refers to data from earlier similar studies such as reference 11 then it should be elaborated in more detail.

Response:

We have elaborated on Foxp1 function as a negative regulator in the discussion and apprised our results in context to other studies. See discussion at line 388-397.

  1. Why were these four AR-target genes selected? Are both Foxa1 and Foxp1 known to bind to their promoters? How are these results in line with the data in reference 11 where it appears that more AR target genes are affected?

Response:

The four genes were chosen from the literature as Fox regulated genes in the context to AR. The data could be extended by RNAseq or a larger panel of AR regulated genes. However, we looked at these genes to assess if Foxp1 or Foxa1 would regulate AR targets in vivo and if this were dependent on testosterone. The data confirms that some selected targets are regulated whereas others are not. Therefore, we are careful in our conclusion, as there could be other genes not negatively regulated but rather positively regulated.

See discussion at line 386-390

Reviewer 2 Report

In the manuscript, the authors investigated the role of FOXP1 and FOXA1 in prostate cancer using an in vivo genetic approach. The experiments are elegant and well conducted, and the results sound well.

I have only a couple of comments for the authors:

-         In results 3.3 and 3.4, the authors performed the knockout of FOXP1 with PTEN knockout and FOXA1 knockout concomitantly with PTEN and FOXA1 knockout. However, as shown in figure 1 and figure 1S, in primary PCa, only 30% of FOXP1 mutations occur concomitantly with PTEN deletion, and no FOXA1 mutation occurs concomitantly with PTEN deletion and FOXP1 mutation. For this reason, the context simulated by the authors could not represent a real PCa scenario. So, how do the authors explain the choice to perform multiple knockouts simultaneously?

-         What pathways are regulated by FOXA1 that could explain the dual role of oncoprotein and tumor suppressor?

Lines 20 and 48: check the spelling of the word “alternated.”

Author Response

We thank the reviewers for the thorough and constructive comments on our manuscript. We have addressed the majority of the questions, which has mainly been clarification and modification of data interpretations. We have added a new supplementary figure containing IHC staining for androgen receptor, as requested by reviewer 2.

We have tried to address all the questions but if any reviewer feels that this has been insufficient, we will happily elaborate on the matter.

For the reviewers information, the text has been edited and moved in the manuscript document. This has resulted in figure texts moved and not standing below the figures. We have not edited this, as the file is with track changes and this would be confusing.

In the manuscript, the authors investigated the role of FOXP1 and FOXA1 in prostate cancer using an in vivo genetic approach. The experiments are elegant and well conducted, and the results sound well.

I have only a couple of comments for the authors:

-         In results 3.3 and 3.4, the authors performed the knockout of FOXP1 with PTEN knockout and FOXA1 knockout concomitantly with PTEN and FOXA1 knockout. However, as shown in figure 1 and figure 1S, in primary PCa, only 30% of FOXP1 mutations occur concomitantly with PTEN deletion, and no FOXA1 mutation occurs concomitantly with PTEN deletion and FOXP1 mutation. For this reason, the context simulated by the authors could not represent a real PCa scenario. So, how do the authors explain the choice to perform multiple knockouts simultaneously?

Response:

            The incident of mutations in human samples shows that concomitantly loss of Foxa1 and Foxp1 rarely occurs in adenocarcinoma but can be found in metastatic prostate cancer. This could suggest that loss of both Fox genes could drive adenocarcinoma to form metastasis. However, it could also indicate that loss of both Fox genes is a disadvantage during prostate cancer initiation. Therefore, we found it interesting to model loss of both Fox genes in the mouse prostate by CRISPR.

Combination with Pten loss is included to accelerate the model by hyper-proliferation and that alteration in the Pten/Pi3K pathway is a hallmark in prostate cancer. The data analysis suggests that loss of Pten can co-occur with loss of Foxp1. In context of Foxa1, the data is difficult to interpret, as loss of Foxa1 is rare in prostate cancer, whereas gain-of-function mutation is more dominant. We included loss of Foxa1 to the study, as we were interested in the interaction between Foxa1 and Foxp1.

We have added this to the discussion section on line 370-372 

-         What pathways are regulated by FOXA1 that could explain the dual role of oncoprotein and tumor suppressor?

Response:

The literature has connected Foxa1 with the AR pathway and upregulation of Foxa1 seems to have an effect during castrations resistance. In contrast, the tumor suppressor function of Foxa1 is less understood but the human data suggests that loss of FOXA1 can occur. Therefore, we seek to assess the loss of Foxa1 during prostate cancer. We have seen dysregulation of App (discussed on line 398-404). However, it is likely that the potential tumor suppressor function of Foxa1 is more complex.   

Lines 20 and 48: check the spelling of the word “alternated.”

We have changed “alternated” to “altered”. We thank the reviewer for pointing this out.

Reviewer 3 Report

In this study, the author showed the role of deletion of FOXP1 and FOXA1 in prostate in vivo model. There are some concerns. Please check the following points.

Major
1. Throughout this study, please add wild type mouse model.
2. The author analyzed the AR pathway. Please analyze AR expression.
3. Loss of FOXP1 increased the proliferation. How about the phenotypical
change?
4. According to previous studies (PMID: 29057879), FOXP1 loss reduced the expression of Akt phosphorylation. However, in this study, the expression of Akt phosphorylation was not changed. Please describe the reason

Minor
1. In the abstract, the description “primer tumors” was found. Please
revise it.
2. In figure 2E and figure 4E, what do the arrows indicate?

Author Response

We thank the reviewers for the thorough and constructive comments on our manuscript. We have addressed the majority of the questions, which has mainly been clarification and modification of data interpretations. We have added a new supplementary figure containing IHC staining for androgen receptor, as requested by reviewer 2.

We have tried to address all the questions but if any reviewer feels that this has been insufficient, we will happily elaborate on the matter.

For the reviewers information, the text has been edited and moved in the manuscript document. This has resulted in figure texts moved and not standing below the figures. We have not edited this, as the file is with track changes and this would be confusing.

In this study, the author showed the role of deletion of FOXP1 and FOXA1 in prostate in vivo model. There are some concerns. Please check the following points.

Major
1. Throughout this study, please add wild type mouse model.

Response:

The control mice used in this study has Pten mutated by CRISPR in the prostate. Each prostate samples have multiple areas with no gene alterations, as AAV containing sgRNA only transduces a subset of the cells. Therefore, areas with normal prostate epithelial are found in each sample and have been used as internal control. Furthermore, we have previously compared the CRISPR model of prostate cancer with wild type mice, mice with injection of non-targeting sgRNA or AAV with only expression of Cre. These mice have all normal prostatic tissues 9 months after injections (Riedel et al., Oncogene 2021). We have commented on this matter on line 78-82 and 223-226.  

  1. The author analyzed the AR pathway. Please analyze AR expression.

Response:

We have added stainings for AR in the prostate tissues, which have been added to Sup figure S4 and mentioned on line 237 and 261.

  1. Loss of FOXP1 increased the proliferation. How about the phenotypical
    change?

Response:

We could not see any clear phenotypical changes. The size and weight were similar between Pten control and Foxp1 and Pten deficient samples. The samples had a larger variation, which compromised statistical analysis. The induction by CRISPR also increased the variation, as some samples had larger areas with edited cells and other samples, only have few areas. We predict that a difference in samples size could have been observed 9 months after initiation but we only followed the mice for 4 months.

  1. According to previous studies (PMID: 29057879), FOXP1 loss reduced the expression of Akt phosphorylation. However, in this study, the expression of Akt phosphorylation was not changed. Please describe the reason

Response:

It is true that decreased levels of pAKT have been observed with loss of Foxp1 in combination with Shq1 mutation. We did not see this but our model did not include loss of Shq1. We addressed the levels of pAKT by IHC or IF, which are not quantifying methods. Furthermore, pAKT staining was performed in the context of loss of Pten, which is a master regulator of pAKT. Therefore, we could not address the relation between Foxp1 and pAKT levels.

Minor
1. In the abstract, the description “primer tumors” was found. Please
revise it.

We have changed that to “prostate cancer”.  

2. In figure 2E and figure 4E, what do the arrows indicate?

We thank the reviewer for pointing this mistake out in the figure text. We have now added the explanation to the respective figure text.

Round 2

Reviewer 3 Report

The manuscript is well revised.